# *“Going Forward Like a Grandmother in the Snow”*: Personal Survival Strategies, Motherhood, and Nature as Resources for Mothers Who Have Experienced Intimate Partner Violence

**DOI:** 10.3390/ijerph20075389

**Published:** 2023-04-04

**Authors:** Minna Sorsa, Hulda S. Bryngeirsdottir, Eija Paavilainen

**Affiliations:** 1Child Psychiatry, Tampere University Hospital, Wellbeing Area of Pirkanmaa, 33520 Tampere, Finland; 2Health Sciences Unit, Faculty of Social Sciences, Tampere University, 33014 Tampere, Finland; 3School of Health, Business and Natural Sciences, University of Akureyri, Solborg v/Nordurslod, 600 Akureyri, Iceland; hulda@unak.is; 4Etelä-Pohjanmaa Hospital District, 60220 Seinäjoki, Finland

**Keywords:** gender-based violence (GBV), intimate partner violence (IPV), help-seeking, nature, maternal health, qualitative research, survival strategies, women’s health, trauma, trauma recovery

## Abstract

After suffering interpersonal violence (IPV), women survivors can access various interdisciplinary services and programmes to guide their recovery. Nevertheless, many vulnerable women postpone seeking help, sometimes indefinitely. Motherhood especially complicates help-seeking because mothers often want to protect both the perpetrator and their children. Understanding women’s resilience, resources, and capacities in surviving IPV, however, could guide the development of helpful services that women actually access. Thus, in our study, we sought to explore the agency, resources, and reinforcing survival experiences of survivors of IPV. Our data, gathered in Finland during the COVID-19 pandemic, consisted of 12 narratives of mothers told in Clinical Ethnographic Narrative Interviews that were subsequently subjected to thematic analysis. Five themes describing personal resources, motherhood, and nature were identified under the overarching metaphor of “going forward like a grandmother in the snow”. Recognising the agency, resources, capacities, and coping mechanisms of women who have suffered IPV can help in developing professional outreach programmes, promoting women’s early access to useful resources, and, in turn, helping them to stop the possible intergenerational transmission of violence.

## 1. Introduction

Women experiencing intimate partner violence (IPV) are often viewed as helpless victims [1,2] who, for one reason or another, cannot leave their perpetrators. However, that negative stereotypical image of learned helplessness has been challenged by the survivor model, which focuses on women’s strengths at the individual and interpersonal levels. According to the model, vulnerable women actively respond to and resist abuse by engaging in help-seeking strategies [1].

Even so, women experiencing IPV may not seek help, often because their perceived failure to meet the community’s expectations for functioning and health fills them with shame, embarrassment, and a sense of being stigmatised in health encounters [3]. Their fear of the perpetrators is another common barrier preventing them from seeking help [4,5], among others widely reported by survivors of IPV [1,4,5,6,7,8]. In Robinson et al.’s systematic review [6], barriers found to discourage formal help-seeking by adult survivors of IPV in the United States included ignorance that their experience was indeed IPV, hindrances in accessing help (e.g., location), fear of the consequences of disclosure, a lack of material resources, personal barriers (e.g., self-blame and a sense of hopelessness), and system failures (e.g., feeling that they are not believed in the formal services). In addition, in their metasynthesis on minority and immigrant women, Hulley et al. [7] identified other barriers related to institutional racism, immigration law, culture, religion, and a lack of diversity in frontline services. Professionals may also characterise survivors of IPV as overcautious but negligent mothers, alienated parents, and/or victims [9], which can further discourage women from accessing adequate support.

Suffering IPV often causes victims to develop various negative feelings about themselves [1,5,8,10,11]. Beyond that, research has shown that children’s exposure to IPV victimisation may impact not only women’s core beliefs but also their actions [1]. Indeed, the situation of mothering while suffering IPV leaves women especially vulnerable. As they try to protect their children from physical and emotional harm [12,13,14,15], they also have to face their perpetrators while in and possibly after having left abusive relationships. From the perspective of safety within such relationships, mothers’ survival has been described as coping with a vulnerable “broken spirit” [14], for the hostility of the situation and the climate of fear may challenge their capacity to protect their children [12].

IPV is one of the risks of intergenerational transmission of violence and/or maltreatment [16,17,18]. All forms of violence can be expected to severely impact children and their lives as adults [19]. According to research on risk and adverse childhood experiences [20], vulnerability exists and can be transferred across generations if the individuals who experience and perpetrate violence do not get the help that they need to end the intergenerational transmission of such behaviour.

Facilitators for help-seeking have also been found in IPV research. Informal support networks can encourage women both to disclose their experiences and to seek appropriate help and treatment. Research has shown that mothers and sisters can facilitate women’s ability to leave, both emotionally and practically. To be able to leave the perpetrator, the women had to overcome their feelings of threat, fear, isolation, and powerlessness [7]. Help-seeking when suffering IPV is connected to social determinants and cultural contexts, since experiences causing vulnerability are formed within culturally constructed life events [3].

At the same time, research on IPV has shown that victims often possess resilience and supportive elements. Survivors of IPV can gain self-respect and hope during their recovery, even in situations with histories involving multiple traumas [11]. Various strategies that help them survive in such demanding situations include persistence and resistance, flexibility, self-care, a belief that they have control over events affecting themselves, and engagement in recreational and/or physical activities and/or hobbies [17]. Women who have suffered IPV have indeed described how their resilience as well as their abilities to survive IPV have developed throughout their lives [15,21]. In an interview study, Vil et al. [2] identified three types of coping: internal (e.g., religion), interpersonal (e.g., leaving the abuser), and external (e.g., relying on others’ support). Coping strategies can also include strengthening connections in nature and thus promote positive emotions, spiritual connections, and positive relationships with oneself and/or counteract negative thoughts [22]. Such strategies following IPV have not been thoroughly studied [23].

Mothers who have survived IPV seem to experience their mothering responsibilities differently from culture to culture. Cultural norms concerning motherhood can restrict such women’s search for help as their social identity and norms of good mothering clash with stigma. The negative feelings imposed by culture or society may discourage women from seeking help in response to IPV [4,24,25]. Even so, mothers have also reported wanting to break the cycle of violence and thereby make their children’s lives better [13,14].

Institutions and policies impact the possibilities for receiving help in cases of IPV [7,26]. Considering that reality, Finnish anthropologist Marja-Liisa Honkasalo [26] has questioned whether acting to counter vulnerability and ensure survival is the responsibility of the individual or society. According to Honkasalo [26], categorising people, e.g., as vulnerable or at risk, creates a system in which individuals labelled as vulnerable may be considered helpless by society, where their personal agency and their ability to act in the world are not recognised. For one reason or another, many women in need of help do not seek it out or face barriers in accessing services that could be helpful for them [1,4,5,6,7,8]. Societal policies, culture, healthcare, and social care systems can also restrict or challenge the exercise of personal agency in society, e.g., by making service options unavailable. When that is the case, people in need are likely to interpret their circumstances as being unavoidable, thus remaining passive in their difficult situations. Against that trend, the same systems could also support such vulnerable individuals in exercising personal agency and resilience [26].

Our research on survivors of IPV in the cultural context of Finland is in line with Honkasalo’s notion of wounded agency [26]. In the study presented herein, we were especially interested in learning about mothers’ resilience, their positive and encouraging capacities, and their challenges with survival while experiencing IPV.

### The Aim of the Study

In our study, we aimed to explore the agency and resources for women with children while experiencing IPV. We inquired into their experiences with survival amid IPV and which resources and capacities have been helpful in those efforts. Such new knowledge might enhance current understandings of intrapersonal ways of coping. Recognising women’s agency, resources, capacities, and coping strategies can be valuable when developing professional outreach programmes seeking to promote their early access to helpful resources.

## 2. Materials and Methods

### 2.1. Study Design

To achieve our study’s aim, we conducted Clinical Ethnographic Narrative Interviews (CENI), a method based on the theory of the benefits and challenges of the narrative approach following IPV and trauma. The CENI method has been developed by Saint Arnault and her international team since 1995 in research on women who have survived IPV and trauma [27,28]. CENIs focus on clinical factors (C), with an emphasis on distress and suffering; ethnographic factors (E), with an emphasis on the cultural and social meaning-making of suffering; and the narrative techniques (N) used when encouraging participants to tell their stories. The purpose of CENIs is to encourage participants to connect various aspects of their lives and examine those aspects from multiple angles. Thus, when CENIs are conducted, four methods are used: social network mapping, body mapping, lifeline construction, and card sorting. Those four methods are the basis for the final step in the process, which involves asking questions about causes, consequences, meaning, the self, and the future. At every turn, CENIs aim to ensure that the survivors being interviewed can tell their stories without being victimised again, while remaining empowered, showing an understanding of themselves, and having their personal capabilities restored [27].

### 2.2. Participants

In our study, we used purposeful sampling, and participation was both voluntary and anonymous. All participants were mothers who identified as having experienced gender-based violence, were older than 18 years old, and were not currently under any known threat of violence. Ultimately, 12 women who had participated in an online survey volunteered for the CENIs. The participants consecutively included in the study were 33–76 years old (M = 49.6). Most had experienced some type of trauma in their lives, including bullying at school and, in time, severe IPV (see Table 1).

### 2.3. Data Collection and Analysis

As part of the CENIs conducted in data collection, each participant completed her own social network mapping, body mapping, lifeline construction, and card sorting, as well as answered questions to guide them in putting all of the pieces together as the final step of the CENIs (see Table 2). Before the interviews commenced, the participants were encouraged to express themselves freely and not to suppress their feelings or reactions but to accept their perceptions of their experiences. Three trained interviewers conducted and recorded all of the interviews, which lasted from 42 to 127 min (M = 94 min). Before analysis, the recorded interviews and all other data created by participants during the interview were encrypted.

The data analysis, conducted by the first author, began with a thorough reading of the interviews. The aim of the first phase of the data analysis was to gain an overview of the interviews’ contents. Then the authors worked together on formulating the research question and the scope of the study. Next, NVivo (QSR) was used to extract data and select text for the purpose of answering the research questions. After that, the data was sorted by selecting quotes reflecting the women’s methods of survival following their experience with IPV. The themes were then further analysed into main themes and subthemes and sorted. In our study, each theme contained an essence, an underlying pattern, or a core concept uniting the women’s experiences [29]. The themes could also contain contrasting or ambiguous perspectives. Next, the themes were subsequently specified, and descriptive chapters about the main themes were written, where the themes were discussed, reflected upon, and verified by the research group. By using original extracts from as many participants as possible, we aimed to help the reader see the connection between the results and the original data. Finally, we analysed the connections between the themes and identified the metaphor that reflected the interconnection of the themes, binding them all together. The phases and aims of the phases of the data analysis are shown in Figure 1.

### 2.4. Research Ethics

The researchers in our study were guided by major principles of research ethics and followed the guidelines of the Declaration of Helsinki [30]. The University of Michigan’s Health Sciences and Behavioral Sciences Institutional Review Board reviewed the study (No. 17.4.2020/HUM00091662), and the Tampere Area Ethical Board issued a statement (No. 15.6.2020/46/2020) about the research project. Ethics and General Data Protection Regulation standards were followed, and all interviewers signed a statement of confidentiality. All participants received an introductory letter with information about the study’s purpose, the research method, and what their participation would involve before they agreed to participate in the study. Before the interviews commenced, they also signed an informed consent form to participate, which emphasised the voluntariness of their participation, their right to withdraw from the research at any time without explanation, the absolute confidentiality of their data, and their anonymity.

## 3. Results

The women described their life histories and experiences of survival as being multifaceted, and an understanding of past incidents and violent events was mingled with the memories of their childhood experiences. Out of the 12 mothers from our study, three had endured severe childhood trauma, five had suffered bullying at school, and seven had sought help for abuse in their adult relationships in the past 12 months (see Table 1). One woman had a refugee background.

The women’s survival episodes had taken place in different everyday situations, which challenged their wounded agency as they reflected on the violence of the situations. The women described their experiences and coping strategies as they unfolded in various challenging circumstances—for example, when their partners were using alcohol or had mental problems, when they left their partners, during subsequent stalking and abuse, when they lived with abusive relatives, and when their abusive past partners lied to social and health care services. The women reported having to endure, overcome, and process a great deal before being able to move on. That goal usually became possible once a new, safe home of their own was available:

“Onhan se oma valinta, kenen kanssa niinku alat suhteeseen, mutta sekin on oma valinta kun sä irtaannut siitä. Eikä sitä ei voi kukaan toinen sanoa että nyt kääri helmat kainaloon ja lähde juoksemaan, vaan että se täytyy tulla iteltä se päätös. Se vaan odottaa sitä oikeeta hetkeä ja niitä voimavaroja. Että saa itelleen sitä voimaa”. (6)

“You choose whom you start a relationship with, but you also choose when to detach yourself from it. Nobody else will come and say, “Hey, pick up your stuff and run away”; instead, the decision has to come from you. You just have to wait for the right moment, the right resources, and the time when you are sufficiently empowered”. (6)

* Numbers in the end of a quote are showing the respondent. Same as below.

As the above quotation shows, the women also discussed their abusive relationships in terms of their own decision-making possibilities within cultural constraints. They stressed that they and their friends and relatives should not have to endure violence, yet violence occurred nonetheless. For the women, leaving their abusive relationships had not been a simple decision made in the moment but a multifaceted battle that had worn down their personal strength. The period was one of uncertainty, when any available resource could make a difference:

“On aikamoinen polku käveltäväks, mutta polkua on vielä eteenpäin. Nyt on se hetki, että nyt ruvetaan paikkaa sitä rikkinäistä ihmistä kasaan sieltä”. (7)

“It is quite a path to walk, but there is still a path ahead. Now is the time to start patching the broken human being back together.” (7).

The personal resources that the women had used in surviving IPV had given them inner strength in the face of life challenges (see Figure 2).

### 3.1. Motherhood and Children as Giving Direction

Several of the participating women’s children had witnessed the physical and/or verbal abuse of their mothers’ abusive partners—that is, the abusive partners’ harassment and/or degrading of their mothers. While defending their mothers from such abuse, some children had been harassed themselves, if not beaten as well. Several of the women pinpointed the mistreatment of their children as being decisive in their actions taken to leave their abusive partners. One woman, who had long remained in an abusive relationship, including throughout pregnancy, realised shortly after giving birth that she would have to end the relationship in order to protect her child:

“Mun piti saada se turvalliseen paikkaan. Ehkä vasta mun tyttären myötä olen alkanut arvostaa ittensäkin että sit jotenkuten, kun mä tajusin etten mä halua mun tyttärelle tällaista. Muistan että pikkuhiljaa aloin miettiä, miksi sitä halusi itselleenkään? Mutta sitä pystyi itse helpommin sietää.” (12)

“I had to get my daughter to a safe place. Only after she was born did I start appreciating myself and realise that I did not want anything like that for her. I remember slowly starting to think, “Why should I even want this for myself?” It had been easier to cope on my own.” (12)

Thus, one participant reported that her child had been her reason for seeking help from formal sources. She and other women realised that, within the complexity of Finland’s health and social care systems, different services focused on different family members. In the end, another woman realised that she was the only one in her family who was not receiving appropriate help. For some of the women, the interventions of child welfare services and the subsequent protection of their children had functioned as a sanctuary.

Many women reported overcoming phases of their abusive relationships and the practical difficulties of detaching from them out of concern and gratitude for their children. The participants reported making decisions solely in consideration of their children and wanting to be good role models who demonstrated to their children that survival is possible. The process required action; however, the women often lacked the capacity and strength for some actions, especially if their abusive partners had used alcohol and/or suffered from mental illness. One woman reported needing to first overcome her sense of helplessness and identify solutions because the only way to avert a real threat to one’s life may be to leave.

Living with small children also helped the participating women concentrate on and enrich their everyday lives instead of focusing on their constant fear of violence. Having small children required them to maintain a steady daily rhythm:

“Kiitollinen itselleni, ylpee itsestäni ku mä sain tehtyy sen, rakkaus omia lapsia kohtaan ja toivo tulevasta. Ei ollu muita vaihtoehtoja.” (7)

“I am thankful for myself. I am proud of myself that I got to [leave an abusive relationship] and of the love I have for my children and the hope I have for the future. There was no other choice.” (7)

The women also described how surviving meant that their children would not suffer in the future, even by forgoing any feelings of revenge and/or hatred towards the abuser that would no doubt grow if their children were to continue enduring such abuse:

“Viimeisillä voimillani mä läksin pääsin sieltä pois. Et jos mä oisin odottanu päivän ni mulla ei ois varmaan ollu voimia lähtee. Et mä oon lähteny sieltä meidän kotoo sillai et mä oon laittanu (lapsen) kärryihin ja lähteny. Ja mul ei ollu mitään muuta.” (7)

“I used the last strength that I had left. If I had waited another day, then I might not have had the strength to leave. I left, put the child in the baby carriage, and left. I had nothing else.” (7)

Amidst an emerging recognition of the need to leave their abusive partners, feelings of love shifted to feelings of hate and fear. The women had tried to conceal from their children the violence that they had endured. Not only did leaving their abusive partners take time, but when the time was finally right, the children had to be involved as well. Participants with older children remarked that, as their children grew, they had increasingly turned against the perpetrator. Even after the abusive relationship had ended, the women continued to consider their children’s perspective and, for example, hoped that the children and the abusive partner could build a relationship irrespective of the abusive past.

“Mun pitää pitää se mun toinen elämä niin piilossa tai häivytettynä lapsilta, että mä jaksan siten että ne ei näe sitä. Tietenkin niillä on niin, että jotkut asiat pitää totuudenmukaisesti ja realistisesti kertoa, mutta sitten tavallaan nii ku se kauna ja ne semmoset negatiiviset tunteet ja vihan tunteet mitä mulla on sitten tota expuolisoa kohtaan ni mä joudun ne sitten häivyttämään.” (2)

“I need to keep that part of my life hidden from the children, and I survive in a way that they don not see it. Of course they have to be told about things truthfully and realistically, but I have to get rid of the resentment and those negative feelings and hatred that I have towards my ex-partner.” (2)

Attending to legal matters concerning former partners and children also required considerable effort and knowledge. The steps taken to advance the separation process were mixed with sorrow and guilt, mostly for what the children had to endure, their loss of trust in adults, and/or the erosion of their self-esteem. Although acknowledging those feelings, professionals in social care services encouraged mothers to change their lives for their children’s sake.

### 3.2. Nature as Calming and Empowering

The women additionally expressed a need to be outdoors in their day-to-day lives. Several women reported currently living near green spaces that made it easy to get outside and exercise in forests or parks. For the women, nature felt safe, and they often ventured outside with their children as well. Nature helped them to endure their bleak circumstances or to regain calmness amidst the difficulty in their lives:

“Toisaalta niitä välineitä, mitkä tietää, että vaikuttaa itseen rauhoittavalla tapaa. Tarkoituksella siis vaikkapa hakeuduin luontoon ja ikään kuin rauhottumaan, kuin “luonnon syliin”. Se, että mä rauhoituin luonnossa, se taas auttoi siihen, että, -mun tavoite oli ikäänkuin, että pysyisin riittävän johdonmukaisena sen suhteen, että mä pääsisin eroon tästä henkilöstä.” (11)

“Such resources, you know, will impact yourself in a calming way. I intentionally went into nature in search of peace, to “go into the lap of nature”. The fact that I can calm down in nature helped me to continue working towards my goal, which is to focus as consistently as possible on getting rid of this person (the abusive partner).” (11)

The women described being in nature as not only calming but also empowering, which had been and continues to be essential for their survival. For instance, enjoying beauty and the fresh scents of nature and recognising the yearly cycle of seasonal changes required the women to concentrate on nature’s most precious details. Two women described how their close observations had encouraged them to reflect on themselves and gave life to new meanings and realisations. Enjoying the beauty of nature was indeed possible when the women focused their energy on their environment—for example, by intently watching the beauty of delicate snow crystals in the winter.

Walking in nature was magical in a sense that it brought about a sense of comfort:

“Mä meen tonne sellaiseen luonto-tai tonne lenkkipolulle mä tiedän et siel on virtaava joki tai järvi on se mikä sitten lohduttaa mua. Järven liike ja se mitä kaikkea siellä on… Ja metsä on se, metsä on, se on semmonen lohdullinen paikka. Että siellä sit näkee niitä, sinne voi purkaa sitä (vaikeita kokemuksia). Se lohduttaa, tulee sellanen hyvä olo.” (7)

“I go into nature or a jogging path, and I know there is a river flowing or a lake nearby, and that is what comforts me. The moving water and everything around it. And the forest is—the forest is such a place of comfort. There, you can see everything clearly, and you can process those difficult experiences. It comforts me and makes me feel good.” (7)

The women considered nature a place of relief and where discomfort was kept at bay. In nature, the women could be themselves without having to justify or undo their actions to others. One woman recalled how she used to spend hours in the forest as a child and always experienced a feeling of ease free of all fear, even in situations when she got lost amongst the trees. Nature felt empowering to her and gave her the strength she needed for everyday life. Another woman who had suffered childhood trauma recalled happy childhood memories of her family fishing and going hiking in the woods.

### 3.3. Being Alone (i.e., Positive Loneliness)

An additional way of surviving the hardships of IPV is by resorting to oneself and one’s inner reflections. One of the participating women described spending her childhood alone at home while her parents worked hard outside the home, but she did not know where they were. As a child, she had not recognised that she was lacking communion with her family, a memory that was now painful in adulthood:

“Ymmärrän nyt, että mä reagoin myöhemmässä elämässä vastaavantyyppisiin tilanteisiin. Siihen, etten mä tiedä mitä tapahtuu, ja mä olen yksin. Mutta sillon mä en on tuntenut sitä turvattomaksi.” (11)

“I now understand how I have reacted later in life in similar situations when I have not known what was going on and have been alone. However, at the time, I did not sense it as insecurity.” (11)

Her hard-working parents had left her to survive by herself as a child, and now as an adult, she had not felt safe disclosing to them that she was suffering from IPV or asking them for help. On the contrary, her painful childhood memories had minimised her trust in others. Surviving on one’s own and maintaining their dignity was expressed as keeping a strong face in front of others. Meanwhile, another woman explained that she avoided meeting others and, in the Finnish winter, even shovelled snow by herself in the middle of the night.

One participant expressed that experiencing the constant threat of violence and unending coercion at home had driven her into isolation from her surroundings and to suppress her own thoughts and ideas:

“Ne (olosuhteet) lamaannutti. Ja kaikki semmonen, ettei ollut mitään omia, ei saanut olla mitään omia toiveita, eikä suunnitelmia eikä mitään semmoista niin kuin, mitään sen kuplan ulkopuolella.” (12)

“Those circumstances were paralysing. And with all of that, you had nothing on your own. You were not allowed to have your own wishes or plans or do anything outside that bubble.” (12)

Although being controlled by another person can feel suffocating, some women report becoming accustomed to it. Some mentioned being too quiet in front of others and how saying no has been difficult. During their violent relationships, loneliness also resulted from the fear that disclosure would only cause more violence. When one woman attempted to leave her violent husband, he began harassing her and threatened to kill her and her parents. Feeling that no help was available and that the risk of homicide was real, she decided to remain silent, take her husband back, and simply try to keep surviving:

“Se vaati multa ihan tosi paljon päivittäistä rohkeutta että koska sithän se meni niin että otin hänet takaisin sen takia että mun ei ois tarvinnut pelätä sitä että hän väijyy jossain nurkan takana ja käy päälle koska mä olen hänet jättänyt. ajattelen että enemmän kuin mitään koska siinä on ollut ihan aidosti se riski eikä mistään oo voinut saada tukee. Ihan kaikki päivittäiset toimet on vaatinut tosi paljon rohkeutta. en mä tiedä mistä mä sen olen sit kaivanut.” (1)

“It took me a lot of courage every day, because I took him back so that I would not have to be afraid of him stalking me around corners and attacking me just because I had left him. I think that there was a real risk, but there was no support available. All of my daily activities required a lot of courage. I do not know where I got it from.” (1)

The woman decided to remain in the violent relationship, which to her seemed to be the best solution possible for overcoming her fears. Ultimately, staying in such a violent relationship did not reduce her fears; hence, she escaped the situation and managed to stay alive. She had felt ashamed for being violated and for hiding her problems. The shame and visible injuries due to the violence discouraged her from meeting with others and opening up to them. By participating in the study, she thus wanted to “break the ice”, so to speak, and give voice to her experience.

Another woman described how she used to visit her mother in silence and never disclose her experience with IPV. Even so, the silent maternal support had felt like “a strong, stony bedrock of security”. Several women described taking similar comfort in solitude and silence:

“Mua auttoi aika. Ihan hirveesti. Mun piti saada olla yksin ja ihan hiljaisuudessa. Se oli semmonen niin voimakas kokemus että se auttoi.” (3)

“Time helped me a lot. I had to be alone in silence. It was such a strong experience that it helped.” (3)

Another woman reported that her choice of social isolation—her way of surviving—had been labelled as “depression”. She had not wanted to succumb to being depressed, for she felt that the diagnosis was a path of no return. To her, reflections on her self-pity seemed to be a negative aspect of loneliness. She therefore advocated solutions such as writing in private and allowing herself to have her own thoughts. Another woman also said that she had been alone but had not felt lonely. For her, being alone seemed to be a positive way of living.

Another woman coped on her own but felt lonely and needed only nature to survive:

“Ei ollut saatavissa apuja sille asialle. Se oli taas sit sitä et sit vaan meni luontoon ja oli yksin ja pähkäili yksin asioita. Siitä vaan oli selvittävä, mentävä eteenpäin.” (7)

“There was no help available for my situation. So, it was again just going into nature, being alone, and contemplating. I just had to survive and move on.” (7)

The women’s loneliness may have also been related to their attempts to prevent friends and/or close relatives from hearing about their suffering at the violent hands of their partners. Several women reported not wanting to burden others and feeling like they could not share their experiences with anyone. Sharing felt uncomfortable, for friends would no doubt strongly advocate leaving their partners. Several women had chosen to stay silent about their experience with violence, in part because others might have disapproved of their choices and would not understand them. Therefore, resorting to isolation had been a choice, one that they could endure.

### 3.4. Activities and Doing Things to Feel Relieved

After one woman noticed that help was unavailable in her social network, she began seeking solutions from outside the network and from the internet:

“Et varmaan se semmonen pieni hiipunut voima (minussa) ja nii ku se että mä kokoajan enemmän ja enemmän luin kaiken mahdollisen mitä oli mitä oli mitään semmoseen huonoon parisuhteeseen ja väkivaltaiseen parisuhteeseen liittyvää, jos oli jossain. Ja tutkin kaikki nii ku turvakodin sivut, kävin lukemassa.” (2)

“There was a small suppressed power (within myself), and all the time I read more and more of all that was written on unhealthy relationships and violent relationships, if there was anything. I searched the webpages of early shelters; I read those.” (2)

For many women, it took time to seek help, and several women reported feeling at some point as though they were frozen or paralysed and as if they had no strength left. One woman had repeatedly watched a movie that gave her hope. Some resorted to including positivity and humour in their everyday lives, while still other means of survival included starting a totally new activity or relearning a skill from childhood—for instance, writing poetry, studying, dancing, swimming, building things, gardening, cooking, knitting, or playing with children:

“Kuitenkin on sitten aikaa harrastuksille jonkin verran. Se on semmonen mulle riittävä ja riittävän hyvä elämä.” (3)

“When there is time for hobbies, then life is sufficient and good enough for me.” (3)

Several women also had pets, such as dogs, that they took out. Favourite pets had been involved in the middle of the strife or shortly after separation, for the perpetrator might have taken the dog to his house without permission. Dogs were a common reason for women to venture outside:

“Sitten noi koirat on tietenkin tossa, kun ne on osa perhettä ja sitten kun tulee se päivän omahetki että on joku oma-aika siinä, niin ne koirat liittyy siihen. Mä lähen sitten niitten kanssa vaikka justiinsa luontoon, metsään, ulkoilemaan ja siinä tulee sitten se semmonen itsestään rauhoittuminen ja asiat saattaa sillä tavalla hyvinkin paljon (kohentua) vaikka ne kuinka hankalalta saattaisi näyttää.” (2)

“The dogs are a part of the family, and when I have a moment to myself, the dogs are involved, as well. I go with them into nature, into the forest, and outdoors in general, and there is such peacefulness, and things seem to improve very much, no matter how difficult it may have initially seemed.” (2)

The animals gave the women a sense of togetherness precisely because it was possible to hug and cuddle with them.

Exercise, such as dancing or jogging, had also brought peace of mind in situations after having received diminishing, mocking, or dismissive feedback from abusive partners. For one woman, going running in the forest felt like she could regain her mental health.

### 3.5. Emerging Will of Making Changes

When focusing on using their strength to survive, the women appeared to focus on taking one day at a time. Many women described how they engaged in various activities as a means of survival. The need to make changes appeared incrementally and raised many questions and points of uncertainty regarding the future, including how one’s life would continue unless another way forward could be found, with the risk of becoming bitter:

“Mä aattelen et sitte ku asiat on takana päin ja ratkastu niin ku mä en sitä tosiaan sitä katkeruutta halua. Ku vedetään kaikki alta. Niin se pettymys tulee ihmisiin ja vähä koko elämään samalla.” (9)

“When things are behind me and solved, I really do not want that bitterness or to have everything taken away. That disappointment sneaks into humans and into all of life at the same time.” (9)

Some participants explained how, having survived IPV, they would no longer worry about trivialities but instead view life from a different perspective. Beyond that, several women described their inner strength as a survival mechanism that originated within themselves. To many, moving on in life meant viewing oneself as a human who could survive irrespective of what they had experienced:

“Se menee sillai, että ei halua ite rämpiä niissä pohjamudissa …mä oon vaan sitten noussu sieltä ja tsempannut itteni aina niihin asioihin, että en mä jää makailemaan ja miettimään. et mä oon itteeni niskasta ottanut kiinni ja sanonut et minähän pärjään.” (5)

“It happens in such a way that you do not want to stumble in the mud. I have always stood up in that situation and encouraged myself to not stay on the ground worrying. I told myself, “I am going to survive”.” (5)

The women’s perspective on survival also meant that they had formed definite opinions about people who would not try to change their lives:

“Mä ymmärrän et jos on semmonen, heikko luonne ja semmonen niin mut emmää sitäkää ymmärrä et sit koko loppuikänsä vaa valitetaan ja ku koko elämä on niin kamalaa.” (7)

“I understand if you have a weak character and if, for the rest of your life, you complain about how your life was just awful.” (7)

The women had also formed opinions on the agency of all women who have experienced IPV, such that their will to change might not have been there from the start. Surviving IPV was explained as an individual process that is forward-thinking and helps women make changes in their lives. The women had tried their best to make their everyday lives functional and identified several specific situations that continued to encourage them to make the necessary changes. For example, their children buoyed their moods, and living and surviving with them on a day-to-day basis both generated and sustained their hope for the future. The women had regained hope in various situations until a more encompassing, life-changing inner motion and will emerged that gave them an entirely new outlook on life. One woman described her changes as a mental and emotional breakdown:

“Mä olin koko elämäni pitänyt kaikki, mä olin aina vahva, mun on ollut pakko olla lasten takia vahva. Nyt sitten mä asuin yksin, koska tytär muutti poikaystävänsä luo. Mä sit asuin siinä yksin ja mä sain romahtaa. Mä siis raahauduin sängystä aamulla sohvalle ja sohvalta sänkyyn illalla. Mä luulen että se on se, että mä romahdin vaan totaalisesti.” (4)

“All of my life, I have thought that I am strong, that I need to be strong for my children’s sake. Then I began living alone because my daughter moved in with her boyfriend. Once I was living alone, I was allowed to break down. I dragged myself out of bed in the morning to the sofa, and from the sofa to the bed in the evening. I think that I just totally broke down.” (4)

All of the women additionally mentioned a time when their experiences with IPV and surviving created a new meaning for their lives and for themselves. In the end, surviving IPV seemed to be able to create a sense of new possibility and direction in life.

### 3.6. “Going Forward like a Grandmother in the Snow”

At the end of the interviews, each woman listed phrases describing the wisdom and tools for survival that had resulted from her dire past experiences. An awareness of the need to move on had developed from an inner discussion, and the pride of having survived despite adversity had made them perseverant and willing to keep going no matter what tried to slow them down—that is, to keep “going forward like a grandmother in the snow” (9).

The women used a Finnish metaphor referring to grandmothers, which can be interpreted as a form of support and as a description of their relationships with the older generation of mothers. The metaphor is set in nature, specifically in the wintry landscape, where movement may be slowed down. Moving forward in the snow is no doubt possible, but only if one is prepared for the circumstances of the environment. In the metaphor, a grandmother moves forward by themselves, in the same way that the women in our study described being alone and finding solitude as sources of strength. The metaphor also describes action within the winter landscape. For our participants, action was needed to survive and overcome life-threatening IPV, and the women had indeed moved forward despite challenges in their lives. During such hardship, perseverance is needed because change may be slow. That reality was expressed by the mothers in our study, all of whom needed all of their inner strength to survive IPV.

## 4. Discussion

### 4.1. Study Results

The women reported that, although their experience with IPV had threatened their agency and ability to act upon the world, they had also found ways to survive and felt resilient in the face of adversity (Figure 2). In the process, they had not been passive objects in seemingly inevitable circumstances in the midst of wounded agency [26], which partly challenges learned helplessness theory [1,2].

Revealing just how multifaceted the connections between surviving and seeking help are, five of the mothers in our study had been outside formal sources of support, whilst seven had been encouraged to seek help from formal sources as a result of motherhood and having children. Several of them had not sought such help, primarily because they lacked the psychological strength to do so. It had taken time for the women to overcome the shame of having endured IPV, which had led them to conceal incidents of abuse within their interpersonal family life. When overwhelmed by cultural norms and stigma, women may limit their searches for help, for the experiences of failure, shame, and guilt can form barriers to help-seeking [4,9,24,25]. Beyond that, professionals may characterise survivors of IPV as overcautious but negligent mothers, alienated parents, and/or victims [9], which can further discourage women from accessing adequate support. The women in our study told narratives from their past, and considering their age at the time of the interview (Table 1), it is possible that supportive formal sources have been developed since those days.

The overarching metaphor of this study was “Going forward like a grandmother in the snow”, which describes the participants’ wisdom and tools for survival, that is, their need to move on, rising from their inner discussion and will to keep on going. This metaphor is in line with the results of two other studies [15,21], where female survivors described the importance of their own resilience and other personal abilities they had developed throughout their lives in surviving IPV [15] and even experiencing post-traumatic growth (PTG) [21]. In our data, the women were fairly well educated (Table 1). In this study, we did not study the impact of education on intrapersonal resources and capacities, yet quantitative studies would be valuable in further exploration of these connections.

The women in our study had several resources for survival available to them, which continued to give them intrapersonal strength as individuals. That dynamic underscores the value of engaging women as partners in the development of suitable approaches within services, which can increase their resilience, as Woods-Jaeger et al. [13] have previously suggested.

Although the participating women were protective of their children, which aligns with the findings of other studies [12,13,14,15], attempts to protect their children from exposure to IPV did not always succeed, and those failures were liable to their broken spirits [14]. Several women described that their children had indeed witnessed violence and suffered negative consequences as a result of IPV. Similar to those children, several participants reported childhood trauma of their own (see Table 1), which also aligns with past findings on the intergenerational transmission of violence and abuse [16,17,18]. In general, our participants described how their childhoods and experiences as young adults had impacted their lives as adults, as Norman et al. have additionally identified [19]. Against those experiences, the women in this study showed intrapersonal strength and wanted to make changes in their lives. Our results thus add to the discussion about breaking the intergenerational cycle of adverse childhood experiences within the context of IPV. Moreover, revealing positive aspects also revealed in past research, the women in our study expressed many forms of resilience [11,17], along with the need to improve their children’s lives [13,14] and cultivate a supportive family environment [16].

In a new study on IPV, nature was described as a calming and empowering element for women. Along those lines, Silva et al. [22] developed a programme in which victims of domestic violence could immerse themselves in natural environments and, as a result, experience positive changes in their self-esteem and quality of life. Moreover, Moore and Van Vliet [23] found that nature served as a source of emotional regulation and spiritual connection, as well as facilitating greater acceptance, reducing dissociation and negative thinking, and increasing attention to the here and now. When the women in Finland in our study spent time in nature, they felt calmer and more capable of coping with the challenges in their lives. Our results thus corroborate past findings from other studies conducted in Finland on nature’s contribution to well-being, especially among youth [31,32]. For instance, Rantala et al. (2020) found that nature has a positive psychological effect on families and young people. In our study, the women enjoyed nature and life outdoors, which, in family studies, has proven to be a factor in strengthening bonds within families [31]. Spending time in nature can also be driven by cultural phenomena. Data collection occurred during the COVID-19 pandemic, when women in Finland increased their outdoor recreational activity [32]. Such crises can indeed drive the desire to connect with nature, which can consequently increase one’s capacity to manage life tasks [32]. Whether actual or simulated in a virtual environment, nature’s role in strategies to survive IPV warrants further research.

The women in our study described how self- and social isolation gave them positive loneliness and solitude without needing to tell anyone about their situation. In such isolation, they could maintain their dignity (keeping a strong face) and realise their desire to protect their relatives and friends from exposure to their suffering. That tendency may relate to the results of Robinson et al. [6], who have described the fear of the consequences of disclosure for partners and family members. In our study, the women wanted to determine their situations on their own. Solitude included a positive aspect stemming from, for example, childhood experiences or silent communication with close relatives. The women also reported strengths in the midst of their feelings of hopelessness and stigma towards IPV survivors. Honkasalo [26] has previously described these negative feelings in society as vulnerability, a wounded agency, where societal systems or cultures restrict or challenge the exercise of personal agency in society. We interpret that the women may have resorted to solitude as an alternative way of enduring their situation, and on the other hand, it may be a Finnish cultural feature that has the aspect of positive loneliness. In this study, we showed that the women possessed resilience and positive endurance at the same time as they were suffering IPV.

The activities described by the participants in our study were pursued as a means of survival, to feel relieved, enrich their lives, and thereby gain personal strength. Their activities also allowed them to escape the challenges of their everyday lives and concentrate on something else in the meantime. The women did not use religion for that purpose, despite the survival strategies documented by St. Vil et al. [2] among black low-income women experiencing IPV, who employed survival strategies such as leaving their abusers, fighting back, and relying on informal and formal sources of support. In our study, by contrast, the women very much relied on themselves. In their cases, recovery could start once they had found a new, safe home for themselves and their children, as also shown in past research [21,33].

For the women in our study, the need to make changes emerged stepwise. They first noticed the need for a change and, in some situations, needed to resolutely focus on such a change. They repeatedly sought solutions to stop the violence and find routes to exit their violent circumstances, as Robinson et al. [6] and St. Vil et al. [2] have similarly found. The culturally dictated shame of staying in a violent relationship was connected with the culture’s attitude that people who do not leave abusive relationships have weak character. Even so, the chief reason for not leaving was not having the strength to endure the process, all of which had been exhausted in the midst of everyday life with children and a manipulative partner. The women’s agency had specific cultural characteristics, as Honkasalo has suggested [26]. In our study, the women relied on their resilience and agency aside from formal sources of help. Action was needed to survive and overcome a life lived amidst IPV, which drained all of their personal strength and required them to use personal resources to move on. As women in Finland, they drew upon a metaphor of grandmothers and past women who persevered and kept going no matter life’s challenges—namely, “going forward like a grandmother in the snow”.

### 4.2. Limitations

We evaluated our research process and the manuscript with reference to the Standards for Reporting Qualitative Research (SRQR) [34], which guided us in all phases of our research, beginning with a concise description of the study topic as a means to develop a qualitative approach. We have covered all 21 SRQR items in our report.

When performing qualitative research, researchers have to rely on the participants’ willingness and ability to discuss their experiences in depth, which could have been a limitation in our study. However, we also recruited women with personal IPV experience who were experts on the topic being studied. It was always the choice of the participants in our study to tell their story in their own words.

Since the CENI guide does not especially focus on the intergenerational transmission of violence, the interviews had to be devised to ensure the participants’ sense of safety by talking about their exposure to trauma without being victimised all over again but by feeling empowered and focusing on their personal capabilities [27]. The participants of this study were all mothers at an older age, which can be a limitation.

The small sample size, the lack of information regarding the participants’ access to social services and legal protection, as well as the limited information about the socioeconomic status of the participants in this study, have led to sample biases that could be seen as limitations. It would be useful to address and meet these limitations in future studies on the subject. However, the aim of this study was not to generalise about the findings but to increase and deepen the understanding of the women’s experience.

Among other limitations, nearly all of our participants were women from Finland, which resulted in a somewhat homogeneous sample, even though we were especially interested in that particular population’s cultural perspective. Along similar lines, the researchers in the study were white women from two Nordic countries with training in healthcare, mental health, and education. Even so, the researchers’ knowledge of the topic enabled them to reflect on the results and gauge their trustworthiness.

The themes in this study describe the survival of IPV victims from the perspective of 12 IPV survivors with a Finnish cultural background. Last, we presented quotations from our participants as verification of the themes identified. We have presented the data analysis process in Figure 1 to enhance the trustworthiness and credibility of the data analysis phases. The sorting of the quotations and the identification of preliminary themes and subthemes were conducted by a single researcher. In the next step, the whole study group scrutinised the results, and two researchers were involved in the identification of the metaphor connecting all themes.

## 5. Conclusions

Considering that self-help and self-care are resources in the search for well-being, the non-disclosure of violence in intimate relationships may be a risk to women and their families. For that reason, it is vital to gain insights into the intrapersonal survival strategies of women who have suffered IPV, expand current understandings of their thoughts, and maintain their dignity. Such insights can be crucial for identifying the best ways to support and help victims of IPV safely exit their violent relationships.

The overarching metaphor of this study was “going forward like a grandmother in the snow”, which describes the participants’ wisdom, perseverance, and tools for survival, that is, their need to resist and move on, rising from their inner discussion and their will to keep on going. Our results reveal that women survivors of IPV in Finland use intrapersonal survival strategies such as going into nature, reflecting on motherhood and their connections with their children, engaging in positive loneliness, engaging in activities, and looking forward to recreating their agency and overcoming their vulnerable broken spirit and/or wounded agency. All this includes their emerging will to make positive changes in their own lives. They associate their survival with their role models, which are the past generation of Finnish mothers who pushed through the snow no matter what. That perspective may be culturally specific to women in Finland and needs further research and scrutiny. Listening to the women’s stories and engaging them in developing appropriate treatment options may be an asset for developing trauma-informed services. Recognising the agency, resources, capacities, and coping strategies of women can be helpful when developing professional outreach programmes so that women can access helpful resources earlier, thus helping them to stop the possible intergenerational transmission of violence. The data collection for this research took place during the COVID-19 pandemic. Since the method of CENI focuses on the life story of each participant, we did not address the effects of the pandemic on their present lives. However, it is important to do so in future research.

## Figures and Tables

**Figure 1 ijerph-20-05389-f001:**
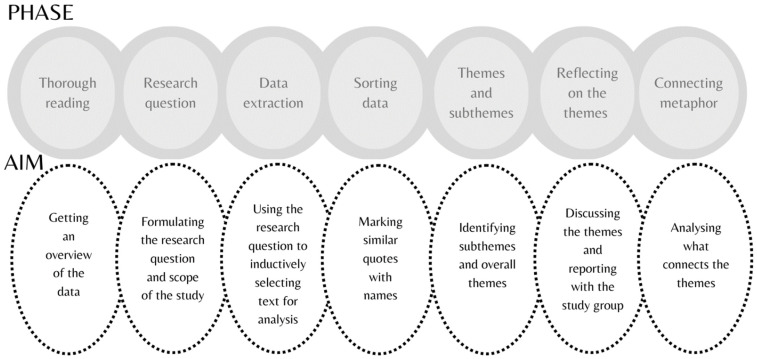
The phases of data analysis and the aim of each step of the analysis.

**Figure 2 ijerph-20-05389-f002:**
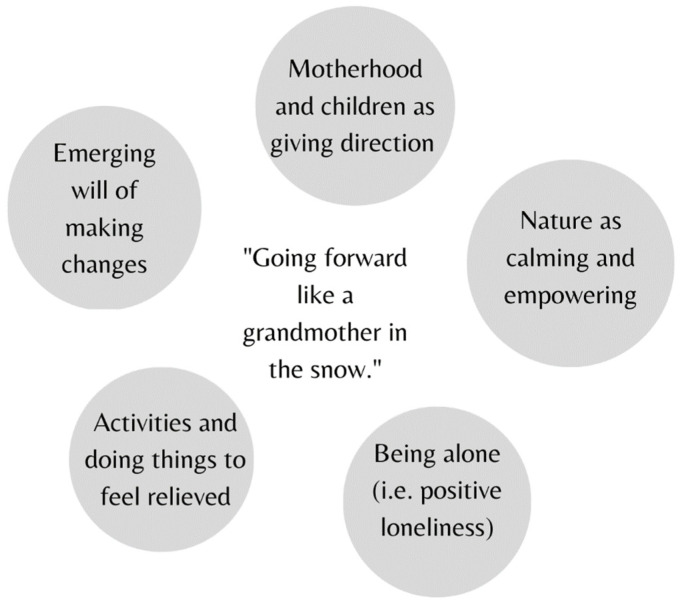
The main resources for surviving IPV, according to female survivors of this study.

**Table 1 ijerph-20-05389-t001:** Background of the participants.

	(n)	%
**Age (years)**		
<45	5	41.7%
46–55	5	41.7%
<56	2	16.7%
**Level of education**		
High school	2	16.7%
Technical or junior college	3	25%
College graduate or candidate degree	3	25%
Master’s degree	4	33.3%
**Children**		
One child	2	16.7%
Two children	6	50%
3 ≤ children	4	33.3%
**Problems, but did not seek help in the past 12 months**		
No	7	58.3%
Yes	5	41.7%
**History and consequences of trauma ^1^**		
Happy childhood	4	33.3%
Childhood with some negative memories	2	16.7%
Traumatic childhood	6	50%
Bullied at school	5	41.7%
Severe IPV	6	50%
Prolonged stalking	3	25%
Traumatic refugee history	1	16.7%
Own child’s or children’s suffering	7	58.3%
Chronic diseases	4	33.3%

^1^ The background information on the consequences of trauma is based on the women’s narratives during CENIs.

**Table 2 ijerph-20-05389-t002:** The basic steps of the CENI interview and the purpose of each step as followed in the present study.

Steps in the Interview Process *	What Was Completed in Our Study	Purpose
Step 1: Social network mapping	The woman drew a map describing her social network while reflecting on her relationships and her feelings towards them and situating herself within her social context. She also considered the value of her social resources and her perceived feelings of negativity and stigma coming from those resources. She regularly returned to the drawing during the interview, added to it, and/or further connected to it.	To frame the woman’s social context and define and understand her support system and social stressors.
Step 2: Body mapping	The woman drew and/or wrote comments on the body map that described the feelings and bodily sensations that she was experiencing in the moment (e.g., pain, numbness, cold, and heat).	To understand the woman’s symptom burden and views on her body in light of trauma.
Step 3: Lifeline construction	The woman constructed her lifeline from childhood to the present in a narrative way by mapping the most relevant past traumatic events (“downs”) and the most relevant happy events (“ups”) in her life.	To organise and link traumatic and happy events and feelings, all to understand the meaning and context of those events and the subsequent development of life.
Step 4: Card sorting	From a stack of cards, the woman chose cards that characterised her low point, each of which described an emotional or physical feeling. She could choose as many cards as she needed. Next, she clustered the chosen cards (i.e., feelings) into relevant groups and explored, discussed, and explained each group while trying to pinpoint the meaning of those feelings.	To define and reflect on the woman’s negative feelings and symptoms due to trauma and the search for meaning, and to indicate her need for help and what kind(s) of help she needs.
Step 5: Questions asked	The interviewer helped the woman interpret her feelings based on the cards that she had chosen by asking her to think about the causes of those feelings, their meaning at the time and now, and how the meaning of her feelings has affected her and her life.Last, the woman connected the meaning of her feelings to her social network drawing, body map, and lifeline. While doing so, the researcher asked her to summarise and describe who she is, her knowledge about her life, and her plans for moving forward.	To assist the woman in making sense of her narrative by interpreting causes, meanings, consequences, and future possibilities.

* [27].

## Data Availability

The data cannot be made publicly available in data repositories because it contains sensitive issues. The data to support the findings are available upon request from the corresponding author.

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
