# Peer review of "“Going Forward Like a Grandmother in the Snow”: Personal Survival Strategies, Motherhood, and Nature as Resources for Mothers Who Have Experienced Intimate Partner Violence"

_ijerph, 2023, doi:10.3390/ijerph20075389_

Round 1

Reviewer 1 Report

This is a fascinating study of an important issue.  I have some concerns, however.  First, the sample size (N=12) is very small and makes rigorous analysis difficult.  Likely because the evidentiary bases is so narrow, the authors treat the respondents as a largely undifferentiated block.  Did class or location or age or health circumstances or family size or access to familial and peer networks affect responses to IPV?  What about access to social services or legal protection?  Did they influence the women's responses to IPV?  Did women from cities, for instance, rely on different coping mechanisms than those residing in more rural communities?  With only twelve cases, the authors cannot systematically explore such potentially significant elements.  Similarly, the authors describe their methodology but do not discuss sample biases.  Were the CENI respondents representative of IPV victims in Finland?  Did the pandemic affect feelings of isolation or their access to support services in important ways?  It seems puzzling that the pandemic is not discussed.  In short, the authors tells a powerful, engaging story of individual stoicism, resilience, and tenacity.  But the analysis of survival strategies, personal choices, and perceived options is underdeveloped.

Author Response

This is a fascinating study of an important issue. 

Thank you for your encouraging feedback!

I have some concerns, however.  First, the sample size (N=12) is very small and makes rigorous analysis difficult.  Likely because the evidentiary bases is so narrow, the authors treat the respondents as a largely undifferentiated block.  Did class or location or age or health circumstances or family size or access to familial and peer networks affect responses to IPV?  What about access to social services or legal protection?  Did they influence the women's responses to IPV?  Did women from cities, for instance, rely on different coping mechanisms than those residing in more rural communities?  With only twelve cases, the authors cannot systematically explore such potentially significant elements. 

Thank you for your feed-back. We show in the background the gaps in current knowledge regarding IPV survivors, and we can show that this type of research has not been conducted previously. In situations, where we look for new knowledge, a qualitative approach can be defended, which explores and describes what is visible and exists in the real world. Our research question was:  “What are the mother’s survival experiences, and which resources and capacities are helpful in surviving IPV?”

The sample size is relevant in qualitative research, and n=12 is a size that has been used in qualitative studies. The large sample size of the study can be seen as a limitation when drawing out the complexities of the phenomenon in detail. Thus, it could be useful in future studies of this phenomenon, to use a smaller sample in order to better elaborate on similarities and differences between participants’ stories.

When talking about trauma exposure and the delicate item regarding IPV, we see that the chosen qualitative data collection and analysis methods were appropriate not to revictimize the women. The methodology was approved by ethical boards both in the US and in Finland, and should be the least intrusive to participants. The area of interest is very difficult to study, and the CENI interview allows women the right to tell their unique story with their own voice. We see the methodology and analysis are line with the chosen theoretical framework. We as researchers have chosen what we wanted to study within the CENI interviews, we acknowledge these decisions, e.g. we have used thematic analysis as an essentialist or realist method, which reports experiences, meanings and the reality of participants. We did not utilize an approach such as discourse analysis or a comparison to society. Themes capture something important about the data in relation to the research question, and represent a patterned response or meaning within the data set. Our methodology was in line with our research question. Our research questions were not class, location, age, health circumstances, family size or access to familial and peer networks affecting IPV. We agree with the reviewer that these, as well as access to care and living within cities or rural areas are all important and interesting topics for research. Yet, this study was a qualitative study with a more narrow focus, than the reviewer might have favoured.

We have added to 4. Limitations: “The aim of the study was to increase the knowledge and deepen the understanding of the phenomenon and not to generalize about the findings.”

Similarly, the authors describe their methodology but do not discuss sample biases.  Were the CENI respondents representative of IPV victims in Finland? 

Thank you for your feed-back. We reviewed “2.2. Participants” (line 141), where we wrote: “In our study, we used purposeful sampling, and participation was both voluntary and anonymous. All participants were mothers who identified as having experienced gen-der-based violence, were more than 18 years old, and were not currently under any known threat of violence. Ultimately, 12 women who had participated in an online survey volunteered for the CENIs. The mothers consecutively included in the study were 33–76 years old (M = 49.6). Most had experienced some type of trauma in their lives, including bullying at school, and, in time, severe IPV (see Table 1).

  • We added that the mothers were consecutively included in the study.

There is not such definition, as “representative of an IPV victim in Finland”. These women are always their own personalities, with their own life experience and life situation. They are women who voluntarily have told their experiences to researchers and want to share them, as usually is in qualitative research. As such, these experiences help the readers understand the life of IPV victims, and bring the reality of these women visible. This hopefully helps to develop services for these women.

Did the pandemic affect feelings of isolation or their access to support services in important ways?  It seems puzzling that the pandemic is not discussed. 

Thank you for your thoughts. In our text, line 603 we mention the pandemia: “Data collection occurred during the COVID-19 pandemic, when women in Finland in-creased their outdoor recreational activity [32]. Such crises can indeed drive the desire to connect with nature, which can consequently increase one’s capacity to manage life tasks [32]. Whether actual or simulated in virtual environment, nature’s role in strategies to survive IPV warrants further research.”

The CENI interviews were focusing on their whole life story (See Table Step 2, line 164): “Lifeline construction” and they talked about their histories of abuse. Being in a current violent situation was an exclusion criteria for the project overall. All women interviewed should have had at least 6 months since moving out from a violent relationship. In the text we wrote (line 143): “who were not currently under threat of violence”.

In short, the authors tells a powerful, engaging story of individual stoicism, resilience, and tenacity.  But the analysis of survival strategies, personal choices, and perceived options is underdeveloped.

Thank you for your feed-back, we think that the article results give one approach to reality, which can benefit the treatment and understanding of what helps the IPV survivors. We think the thematic analysis follows the focus of the study, and the study responds to our research questions. We have added to limitations (line 668): “The themes in this study describe the survival of IPV victims from the perspective of 12 IPV survivors within a Finnish cultural perspective.” We agree that there are many unattended areas in need of further research.

The article was edited via “Edit my English”, and we hope the whole article now reads better.

Reviewer 2 Report

This paper discusses an important topic. The methods are fascinating and the findings are illuminating. The findings could have important implications for development of trauma-informed services for women who have experienced IPV, including the incorporation of outdoor activities as a means to reduce stress. I commend the authors for doing this meaningful work, and I am happy to recommend that this article be published with minor revisions. 

There are some significant issues with the English prose throughout the paper that should be addressed.

For example, in line 36, it states: "the failure to meet community-held expectations of functioning and health cause shame..." This is a subject/verb agreement error. The subject (failure) is singular, and so the verb should be causes, not cause. 

There is another example on line 71: "vulnerability exists and transfers across generations if the parties of violence do not get the help for breaking the intergenerational chain of violence." Perhaps there are some words missing? In my eyes, a more sensible version of this sentence might read: "... if those who experience and perpetrate violence do not get the help they need to end the intergenerational transmission of violence." 

On line 89, it states: "mothers may experience failure, shame, and guilt..." In the context of this paragraph, it seems that the authors may be referring to feelings of failure rather than failure itself. 

On line 96, it states "She explicates that grouping people e.g. as vulnerable and classifying them as risk creates a system, where the individuals can be controlled by society." This should probably read: "classifying them as at risk..." Further, the comma after "system" should not be there since what follows is a dependent clause. There is a similar comma error on line 99.

Line 101 states: "It is noteworthy, whether the societal policies and systems restrict, challenge or enhance to use the own agency and personal resilience, or whether such systems force a person to remain a passive object in unavoidable circumstances." The bolded text is sufficiently grammatically problematic that I am actually unsure of its intended meaning. 

The very next sentence appears to have a typo in it: "The “wounded agency” [26] of IPV survivors is of interest to use, as we will study IPV survivors in the cultural context of Finland." 

These writing issues persist throughout the rest of the paper, and include an error in the main figure (Figure 1). "... doing things to feel relieve..." This should read "relief" or "relieved." 

In most instances the meaning of the writers is clear, but there are frequent instances where the readability and precision of the language in the text could be significantly improved. There are even some instances where the prose issues obscure the intended meaning. Because of how important this work is, it would be a shame if these easily fixed writing issues were to detract from the message of the paper. I strongly recommend a close prose edit for English grammar and writing style. 

Author Response

This paper discusses an important topic. The methods are fascinating and the findings are illuminating. The findings could have important implications for development of trauma-informed services for women who have experienced IPV, including the incorporation of outdoor activities as a means to reduce stress. I commend the authors for doing this meaningful work, and I am happy to recommend that this article be published with minor revisions.

Thank you for your encouraging feedback!

There are some significant issues with the English prose throughout the paper that should be addressed.

Thank you, we have sent the English for editing via “Edit my English”, and hope the whole article now reads better.

For example, in line 36, it states: "the failure to meet community-held expectations of functioning and health cause shame..." This is a subject/verb agreement error. The subject (failure) is singular, and so the verb should be causes, not cause.

Thank you, we have corrected this to “causes”.

There is another example on line 71: "vulnerability exists and transfers across generations if the parties of violence do not get the help for breaking the intergenerational chain of violence." Perhaps there are some words missing? In my eyes, a more sensible version of this sentence might read: "... if those who experience and perpetrate violence do not get the help they need to end the intergenerational transmission of violence."

Thank you, we have corrected this sentence to the one suggested by the reviewer.

On line 89, it states: "mothers may experience failure, shame, and guilt..." In the context of this paragraph, it seems that the authors may be referring to feelings of failure rather than failure itself.

Thank you, we have corrected this to “feelings of failure”.

On line 96, it states "She explicates that grouping people e.g. as vulnerable and classifying them as risk creates a system, where the individuals can be controlled by society." This should probably read: "classifying them as at risk..." Further, the comma after "system" should not be there since what follows is a dependent clause. There is a similar comma error on line 99.

Thank you, we have corrected this to “classifying them as at risk”. We removed the commas after system and on line 99.

Line 101 states: "It is noteworthy, whether the societal policies and systems restrict, challenge or enhance to use the own agency and personal resilience, or whether such systems force a person to remain a passive object in unavoidable circumstances." The bolded text is sufficiently grammatically problematic that I am actually unsure of its intended meaning.

Thank you, we agree that the sentence and meaning is difficult, so we have corrected the sentence to read (line 98-108):  

“According to Honkasalo [26], categorising people e.g. as vulnerable and as at risk, creates a system in which individuals labelled as vulnerable may be considered helpless by society, where their personal agency and their ability to act in the world is not recognized. For one reason or another, many women in need for help do not seek for it or face barriers in accessing services that could be helpful for them [1, 4–8]. Societal policies, culture, healthcare and social care systems can also restrict or challenge the exercise of personal agency in society, e.g. by unavailable service options. When that is the case, people in need are likely to interpret their circumstances as being unavoidable, thus remaining passive in their difficult situations. Against that trend, the same systems could also support such vulnerable individuals in exercising personal agency and resilience [26].

The very next sentence appears to have a typo in it: "The “wounded agency” [26] of IPV survivors is of interest to use, as we will study IPV survivors in the cultural context of Finland."

Thank you, we corrected the sentence to read: “Our research on survivors of IPV in the cultural context of Finland, is in the line with Honkasalo’s notion of wounded agency [26].

These writing issues persist throughout the rest of the paper, and include an error in the main figure (Figure 1). "... doing things to feel relieve..." This should read "relief" or "relieved."

Thank you, we have corrected this to “doing things to feel relieved”, in Figure 1 and throughout the text. We also edited other names of themes and the main metaphor according to the suggestions from the English editing service.

In most instances the meaning of the writers is clear, but there are frequent instances where the readability and precision of the language in the text could be significantly improved. There are even some instances where the prose issues obscure the intended meaning. Because of how important this work is, it would be a shame if these easily fixed writing issues were to detract from the message of the paper. I strongly recommend a close prose edit for English grammar and writing style.

Thank you, we have sent the English for editing via “Edit my English”, and we hope the whole article now reads better.

Round 2

Reviewer 1 Report

The authors sharply the language and narrative, which improves the essay.  They defend their analysis, however, more than they engage my suggestions or strengthen the contribution.  While the shortcomings are not addressed and I see missed opportunities to enhance the scope and significance of the analysis, the paper is nonetheless interesting and thoughtful.

Author Response

Point-by-point response to reviewers

All major changes to the manuscript are in red

Reviewer 1

Open Review

(x) I would not like to sign my review report

( ) I would like to sign my review report

Quality of English Language

( ) English very difficult to understand/incomprehensible

( ) Extensive editing of English language and style required

( ) Moderate English changes required

(x) English language and style are fine/minor spell check required

( ) I am not qualified to assess the quality of English in this paper

Yes                    Can be improved                     Must be improved                          Not applicable

Does the introduction provide sufficient background and include all relevant references?

( )                      (x)                     ( )                      ( )

Are all the cited references relevant to the research?

(x)                     ( )                      ( )                      ( )

Is the research design appropriate?

( )                      (x)                     ( )                      ( )

Are the methods adequately described?

( )                      (x)                     ( )                      ( )

Are the results clearly presented?

(x)                     ( )                      ( )                      ( )

Are the conclusions supported by the results?

( )                      (x)                     ( )                      ( )

Comments and Suggestions for Authors

The authors sharply the language and narrative, which improves the essay.  They defend their analysis, however, more than they engage my suggestions or strengthen the contribution.  While the shortcomings are not addressed and I see missed opportunities to enhance the scope and significance of the analysis, the paper is nonetheless interesting and thoughtful.

The authors´step by step response to the reviewer’s comments from the reviewer:

The authors sharply the language and narrative, which improves the essay.

Thank you, we are so pleased to hear that.

They defend their analysis, however, more than they engage my suggestions or strengthen the contribution. 

Thank you so much for this important comment. We have now responded in the best way we can to your useful suggestions, which we find very beneficial to strengthen our contribution. However, we are happy to receive feedback on the items needing improvement. We have now utilized your very helpful 1st phase review comments.

While the shortcomings are not addressed and I see missed opportunities to enhance the scope and significance of the analysis.

Thank you for pointing this out. We have now made greater efforts on addressing the shortcomings that you have pointed out. We have made some changes in Introduction, Discussion, and Conclusions. Also, we now better explain the analysis of the research data, (see Figure 1)

The paper is nonetheless interesting and thoughtful.

Thank you so much for your kind words and useful comments and suggestions. We hope that our response to your review have made the paper even more interesting and thoughtful ?

We have now responded more thoroughly to your former review as well as your comments in this second round of review, in following ways:

  • Minor changes in Introduction, Discussion and Conclusions
  • We have now added to the chapter of Limitations, where we explain that the small sample size (n=12), the lack of information regarding access to social services and legal protection, as well as limited socioeconomic information of the participants can serve as a limitation to this study.
  • We now describe the step to step phase of the data analysis of this study, using text and a figure (Figure 1)
  • We have now addressed the possibility of sample biases in the chapter of Limitations
  • In the chapter of Conclusions, we now explain why we don’t address the effects of the COVID-19 pandemic on the participants’ lives, even though the data collection took place during the pandemic.
  • In order to improve and support the conclusions better with the results, we have now added some text both to the results and to the conclusions.

We emphasize that all major changes in the manuscript are in red.
